# Evaluation of 5-[(Z)-(4-nitrobenzylidene)]-2-(thiazol-2-ylimino)-4-thiazolidinone (Les-6222) as Potential Anticonvulsant Agent

**Mariia Mishchenko [1]**, **Sergiy Shtrygol' [1]**, **Andrii Lozynskyi [2]**, **Mykhailo Hoidyk [2]**, **Dmytro Khyluk [3]**, **Tatyana Gorbach [4]** and **Roman Lesyk [2,5,*]**

1  Department of Pharmacology and Pharmacotherapy, National University of Pharmacy, 53 Pushkinska, 61002 Kharkiv, Ukraine
2  Department of Pharmaceutical, Organic and Bioorganic Chemistry, Danylo Halytsky Lviv National Medical University, 69 Pekarska, 79010 Lviv, Ukraine
3  Department of Organic Chemistry, Medical University of Lublin, Aleje Racławickie 1, 20-059 Lublin, Poland
4  Department of Biological Chemistry, Kharkiv National Medical University, 4 Nauky Ave, 61022 Kharkiv, Ukraine
5  Department of Biotechnology and Cell Biology, Medical College, University of Information Technology and Management in Rzeszow, Sucharskiego 2, 35-225 Rzeszow, Poland
*  Correspondence: dr_r_lesyk@org.lviv.net; Tel.: +38-(032)-275-59-66

**Abstract:** It was determined that the studied 5-[(Z)-(4-nitrobenzylidene)]-2-(thiazol-2-ylimino)-4-thiazolidinone (Les-6222) affects the cyclooxygenase pathway of the arachidonic acid cascade, the markers of damage to neurons on models of PTZ kindling. In the model of chronic epileptogenesis in mice (pentylenetetrazole kindling), a 4-thiazolidinone derivative showed high anticonvulsant activity, which is weaker than the effect of sodium valproate and higher than Celecoxib. The mentioned compound has a pronounced anti-inflammatory effect in the brain on the background of the PTZ kindling, reliably inhibiting COX-1 and COX-2. The predominant inhibition of COX-2 by 44.5% indicates this enzyme's high selectivity of Les-6222. According to the molecular docking study results, the studied compound revealed the properties of COX-1/COX-2 inhibitor and especially 5-LOX/FLAP. The decreasing content of 8-isoprostane in the brain of mice of the Les-6222 group indicates a beneficial effect on cell membranes in the background of oxidative stress during the long-term administration of PTZ. In addition, Les-6222 significantly decreased the content of neuron-specific enolase, indicating neuroprotective properties in the background of chronic epileptogenesis. The obtained results experimentally substantiate the feasibility of further developing Les-6222 as a promising anticonvulsant agent.

**Keywords:** antiepileptic drugs; thiazolidinones; pentylenetetrazole kindling; inflammation; molecular docking

## 1. Introduction

According to the World Health Organization's data, the prevalence of epilepsy in the world population is about 0.5–1% [1]. In 20–40% of cases [2,3], it is not possible to achieve control over epileptic attacks by standard methods of treatment, which significantly worsens the quality of life of patients, increases economic costs, and creates a difficult choice in terms of the optimal treatment to reduce the frequency of attacks. This explains the urgency of creating new anticonvulsant agents and finding new targets of influence on the pathogenesis of epilepsy.

An evaluation of the effectiveness of new anticonvulsant agents at the stage of pre-clinical studies was carried out on chemo- and electrically induced seizure models in animals. Kindling models of epilepsy are of great importance when an irritating factor in a subthreshold dose repeatedly affects the motor neurons of animals, after which the brain

can generate epileptic discharges without stimulation. As a result, after a specific time, convulsive attacks appear without the influence of a provoking factor. Such animal models of chronic epileptogenesis are close to human clinical pathology [4].

The pathophysiological mechanisms underlying the occurrence and recurrence of epileptic seizures remain poorly understood. Animal models of epilepsy reproduce the dynamic changes that occur in the brain during epileptogenesis (the process that leads to the onset and progression of the disease) and the neurophysiological modifications that underlie ictogenesis (seizure generation and recurrence). These models showed that some mechanisms contribute to the development of both phenomena. Among these mechanisms is neuroinflammation, a complex reaction that includes the release of proinflammatory cytokines and chemokines with glia activation. The content of proinflammatory cytokines and chemokines increases in blood serum, cerebrospinal fluid, and brain tissue of patients with epilepsy [5]. In animals with epilepsy models, their level increases in various brain structures, including the cortex and hippocampus [6].

Glial cells such as astrocytes and microglia are responsible for releasing inflammatory cytokines and chemokines. The activation of astrocytes and microglial cells is the main pro-inflammatory pathway in epilepsy. Therefore, neuroinflammation contributes to the development and progression of epilepsy and can be considered a potential target for treating seizures of various etiologies [7]. Activation of the cyclooxygenase pathway of the arachidonic acid cascade plays a significant role in the pathogenesis of neuroinflammation, and inhibitors of this pathway—non-steroidal anti-inflammatory drugs—are considered as means of adjuvant therapy for epilepsy [8].

Oxidative stress initiates the widespread death of neurons. A vicious circle is then formed in which a cascade of interconnected reactions can be traced. Traumatic damage to neurons promotes the formation of excitatory neurotransmitters, the deficiency of macroergic substances, and the accumulation of calcium ions, nitric oxide, pro-inflammatory cytokines and other substances, which in combination contribute to strengthening the lipoperoxidation process [9,10]. At the same time, active radicals destabilize the function of cell membranes and thereby accelerate the degradation of lipids, contributing to the excess supply of glutamate, calcium ions, and other altering components through microdefects inside the cell [11]. The content of 8-isoprostane, a product of peroxide oxidation of arachidonic acid, allows for assessing the level of oxidative stress with a sufficient degree of reliability and reproducibility of research results, and its amount is directly proportional to the level of free radicals formed [10].

NSE (neuron-specific enolase) is an enzyme of the glycolytic chain found mainly in neurons and neuroendocrine cells of the nervous system. As a result of the damage to brain cells, there is an increase in the level of neurospecific enzymes and their isoforms in the extracellular environment. Therefore, the severity of structural and functional disorders of biomembranes in the CNS can be determined by the degree of NSE increase in brain tissue. Due to the destruction of brain cells, the flow of NSE into the blood increases. In studies devoted to ischemic strokes in adults, a correlation was found between an increase in the level of NSE in the blood and the severity of neurological deficits [12]. Increased content of NSE in adult patients with epileptic syndrome and the dependence of NSE content on the frequency of seizures was also found [13].

The thiazolidinone derivatives have undergone significant development in medicinal chemistry over the past 20 years, including as promising anticonvulsant agents [14–16]. Recently, we designed and synthesized 5-[(*Z*)-(4-nitrobenzylidene)]-2-(thiazol-2-ylimino)-4-thiazolidinone Les-6222 (Figure 1), which has a significant anticonvulsant activity [17,18] and low toxicity level [19].

**Figure 1.** Chemical structure of compound Les-6222.

The current work is devoted to the evaluation of the anticonvulsant activity of Les-6222 on the pentylenetetrazole kindling model and to determining its effect on the content of cyclooxygenase 1 and 2 types (COX-1, COX-2), prostaglandins (PG) E2, F2$\alpha$, I2, thromboxane (TX) of B2, 8-isoprostane and NSE in mouse brain homogenate compared to the classical anticonvulsant sodium valproate and the nonsteroidal anti-inflammatory drug Celecoxib, as well as to reach the actual and predicted effects of Les-6222 on the cyclooxygenase pathway of the arachidonic acid cascade by molecular docking.

## 2. Materials and Methods

### 2.1. Pharmacology Assay

#### 2.1.1. Animals

The experiments were conducted on random-bred male albino mice weighing 20–27 g purchased from the vivarium of the Central Research Laboratory of the Educational and Scientific Institute of Applied Pharmacy of the National University of Pharmacy, Kharkiv, Ukraine. Animals were randomly divided into five groups: intact control; control pathology (kindling without treatment); mice, which were injected with Les-6222, sodium valproate, and Celecoxib, which has anticonvulsant properties on the background of kindling [8]. All procedures performed in studies involving animals were in accordance with the ethical standards of the institution or practice at which the studies were conducted and were approved by the Local Ethical Committee at the National University of Pharmacy, Kharkiv, Ukraine (Approval No: 3/2019).

#### 2.1.2. PTZ-Induced Kindling

PTZ-induced kindling was performed using pentylenetetrazole (PTZ) at a dose of 30 mg/kg intraperitoneally for 16 days [20]. The convulsant was administered simultaneously once a day after each animal was continuously observed for 30 min. The anticonvulsant activity was assessed daily by the following indicators: the day of the first convulsion, the percentage of mice with seizures in each group, the number of seizure-free days, and the severity of the seizures [20]. The drugs were administered 30 min before pentylenetetrazol. The Les-6222 was administered to animals in an effective anticonvulsant dose of 100 mg/kg [18] intragastrically in the form of a suspension stabilized by Tween-80 in a volume of 0.1 mL per 100 g of body weight. The anticonvulsants, sodium valproate (Depakin, Sanofi-Aventis, Ambarès-et-Lagrave, France) at 300 mg/kg and Celecoxib (Celebrex, Pfizer, New York, NY, USA), were used as anti-inflammatory agents at a dose of 4 mg/kg in the form of a suspension stabilized by Tween-80 were used as comparison drugs. The doses of sodium valproate and Celecoxib were chosen based on literature data [8] and our previous reports [21]. Animals of intact control and control pathology groups received purified water intragastrically in a similar volume (0.1 mL per 10 g of animal weight).

#### 2.1.3. Immunochemical Studies

Immunochemical methods examined the brains of mice after PTZ-induced kindling. On the 16th day of the experiment, 1 h after administering drugs and tested substances, the animals were euthanized by dislocation of the cervical vertebrae [22]. The brain was

immediately removed, frozen with liquid nitrogen, stored in a freezer at $-70\ ^\circ$C and homogenized immediately before testing the sample. In the homogenate of the brain of mice, the following was determined using standard species-specific kits: the content of COX-1 (Cyclooxygenase-1, ELISA Kit), COX-2 (Cyclooxygenase-2, ELISA Kit), prostaglandins: PGE2 (Prostaglandin E2 (PG-E2), ELISA Kit), PGF2$\alpha$ (Prostaglandin F2alpha (PGF2alpha), ELISA Kit), PGI2 (Prostacyclin, ELISA Kit), TXB2 (Thromboxanes B2, ELISA Kit), 8-isoprostane (8-isoprostane ELISA, Enzyme immunoassay for the quantitative determination of 8-isoprostane) and NSE (Mouse Neuron-specific enolase (NSE) ELISA Kit).

## 2.2. Molecular Docking

AutoDock Vina v.1.2.0 was used for the molecular docking study. The MM+ molecular mechanic's method optimized the molecular structure, achieving an RMS gradient of less than 0.1 kcal/(mol Å). The basis of the mentioned process is comparing small molecules with the active center of the receptor to identify an imaginary ligand with the greatest affinity [23]. The semi-empirical quantum chemical method PM3 carried out the final minimization of the energies of the studied intermediates until the RMS gradient was less than 0.01 kcal/(mol Å). Verifying the selected docking parameters was performed by re-docking the original ligands from the enzyme spectra and comparing the actual and predicted positions of the ligands inside the allosteric centers. Visualizer Discovery Studio was used to visualize and interpret the obtained data.

## 2.3. Statistical Analysis

For statistical analysis, Statistica 12.0 for Windows was used. Data are reported as the mean $\pm$ standard error of the mean (mean $\pm$ SEM). Statistical differences between groups were analyzed using the parametric Student's *t*-test in cases of the normal distribution; non-parametric Mann–Whitney U-tests in its absence. For the results in the alternative form, Fisher's angular transformation was used. Spearman's rank correlation coefficient was used to identify the relationship between individual indicators [24]. The level of statistical significance was considered as $p < 0.05$.

## 3. Results

The PTZ-induced kindling model allows for studying the effect of drugs and tested thiazolidinone derivative on the convulsive state, which is closest to the real pathophysiological and clinical features of epileptogenesis [20] by repeated stimulation of subthreshold intensity, which causes focal convulsive discharges and generalized convulsive attacks [25].

The results of the study are shown in Table 1. On the 4th day of the experiment, sub-threshold doses of PTZ led to a gradual increase in convulsive activity: the appearance of clonic seizures was observed in the control pathology, Celecoxib and Les-6222 groups. The first paroxysms were recorded in the sodium valproate group on the sixth day of PTZ administration. No statistically significant difference was found between the studied and the control pathology groups in the latent period duration.

The studied Les-6222 showed an anticonvulsant effect, which indicates a statistically significant difference in the number of animals with seizures from the 8th to the 16th day of the experiment. On days 8 and 9 and from 11 to 13, no seizures were registered in animals of the Les-6222 group, which is significantly ($p < 0.05$) different from the parameters of the control pathology group and Celecoxib, in which seizures were observed daily. In the sodium valproate group, which was not inferior to the compound Les-6222, there were no seizures for 8, 10, and 12 days.

**Table 1.** Effects of the Les-6222, sodium valproate and Celecoxib on the course of pentylenetetrazole kindling in mice (M ± m).

| Experimental Data | Group of Animals | | | |
|---|---|---|---|---|
| | Control Pathology (Pentylenetetrazole), *n* = 9 | Sodium Valproate, *n* = 8 | Celecoxib, *n* = 7 | Les-6222, *n* = 9 |
| Dose, mg/kg | 30 | 300 | 4 | 100 |
| Latent period of seizures, days | 4 | 6 | 4 | 4 |
| % of mice with convulsions, the average value for the entire period | 29.83 ± 7.22 | 9.08 ± 3.44 | 29.84 ± 6.71 | 7.29 ± 2.79 |
| Number of days with seizures | 13/16 | 8/16 | 13/16 | 7/16 |
| Number of days without seizures | 3/16 | 8/16 | 3/16 | 9/16 |

On the last day of the experiment (on the 16th day of administration), 77.78% of the animals had pronounced seizures in the control pathology group. In contrast, in the Les-6222 group, only 33.33% of the mice had paroxysms, which statistically confirms the effectiveness of the study compound ($p < 0.01$). According to this indicator, this group was also not inferior to sodium valproate (25%). At the same time, on the background of Celecoxib, convulsions were observed in 85.71% of animals ($p < 0.05$) against the Les-6222 group.

Thus, the effectiveness of Les-6222 is confirmed by a statistically significant reduction in the percentage of animals with paroxysms against the control pathology indicator, which is statistically different from the parameter of the Celecoxib group. No significant differences were found with the sodium valproate group.

According to the number of animals with convulsions, Celecoxib did not show a protective anticonvulsant effect. At the same time, sodium valproate reduced the number of mice with convulsions by 20.75%, which statistically exceeds the corresponding indicator of the control pathology groups ($p < 0.05$) and Celecoxib ($p < 0.01$).

In the Les-6222 group, the proportion of animals with seizures during the experiment was 22.54% less than in the PTZ-induced kindling group ($p < 0.05$) and 22.55% less than in the Celecoxib group ($p < 0.01$). No statistically significant differences were found between the Les-6222 and sodium valproate groups.

Summarizing the dynamics of the development of the convulsive syndrome, it is worth noting that only in the groups of the Les-6222 and sodium valproate a significant decrease ($p < 0.05$) in the number of days without seizures was observed (Table 1). Thus, a high anticonvulsant activity of studied 5-[(Z)-(4-nitrobenzylidene)]-2-(thiazol-2-ylimino)-4-thiazolidinone was found in the PTZ-induced kindling model, which reproduces the conditions of chronic epileptogenesis secondary generalized seizures.

The results of the study of indicators of the cyclooxygenase pathway of the cascade of arachidonic acid, 8-isoprostane and NSE against the background of PTZ-induced kindling are presented in Table 2.

**Table 2.** Effects of Les-6222, sodium valproate, and Celecoxib on the cyclooxygenase pathway of the arachidonic acid cascade, 8-isoprostane and NSE on the pentylenethazole kindling model in mice, M ± m.

| Group | Dose, mg/kg | COX-1, pkg/g of Tissue | COX-2, ng/g Tissue | PGE2, pkg/g of Tissue | PGF2a, pkg/g of Tissue | PGI2, ng/g of Tissue | TXB2, pkg/g of Tissue | 8-isopro-stane, nM/g | NSE, ng/h |
|---|---|---|---|---|---|---|---|---|---|
| Intact control | – | 794.86 ± 4.59 | 132.16 ± 3.44 | 704.70 ± 4.57 | 988.79 ± 15.15 | 6.15 ± 0.05 | 131.66 ± 1.32 | 19.65 ± 0.23 | 4.04 ± 0.05 |
| Control pathology | 30 | 893.06 ± 7.50 | 216.12 ± 7.98 | 407.82 ± 3.08 | 1258.98 ± 18.42 | 3.05 ± 0.06 | 263.44 ± 1.14 | 54.67 ± 1.94 | 70.84 ± 1.01 |
| Les-6222 | 100 | 843.32 ± 9.35 | 119.94 ± 2.48 | 515.04 ± 3.44 | 1030.58 ± 25.39 | 3.88 ± 0.06 | 148.55 ± 2.78 | 45.75 ± 0.93 | 57.83 ± 0.90 |
| Celecoxib | 4 | 778.25 ± 7.73 | 213.25 ± 3.38 | 609.10 ± 3.64 | 1129.62 ± 16.11 | 2.91 ± 0.05 | 183.93 ± 2.43 | 40.42 ± 0.63 | 68.12 ± 0.48 |
| Sodium valproate | 300 | 760.40 ± 8.55 | 103.31 ± 4.33 | 629.42 ± 26.77 | 1085.49 ± 15.47 | 4.11 ± 0.10 | 142.10 ± 1.65 | 32.04 ± 0.54 | 34.44 ± 0.51 |

In the group of control pathology, an increase in inflammatory markers was observed, which indicates the development of a neuroinflammatory reaction against the long-term

administration of the convulsant PTZ. As is known, as a result of the activation of cytokines from vascular endotheliocytes, excessive amounts of secondary messengers (nitric oxide, prostaglandins, etc.) are released and enter the CNS, which causes damage to the brain cells. COX-1 content increased by 12.4%, the COX-2 expression increased almost 2-fold, PGF2$\alpha$ content increased by 27.3%, TXB2 increased by two-fold, and PGE2 and PGI2 production decreased by 42.1% and 50.4%, respectively ($p < 0.01$).

Despite belonging to highly selective COX-2 inhibitors and lipophilicity, which determines the ability to cross the blood–brain barrier, Celecoxib did not affect the increased level of this COX isoform in the brain (Table 2). However, the increased content of COX-1 due to the effect of Celecoxib decreased to the level of the intact control, the content of PGE2 increased by 49.4%, PGF2$\alpha$ decreased by 10.27%, the level of PGI2 was stable, and TXB2 decreased by 30.2% compared to the corresponding indicators of the control group pathologies.

Sodium valproate significantly ($p < 0.01$) affected all the investigated markers of the state of the arachidonic acid cascade compared to the indicators of the control pathology group: the level of COX-1 decreased by 14.9%, COX-2—by 52.2%, the content of PGE2 increased by 54.3%, PGF2$\alpha$ decreased by 13.8%, PGI2 increased by 34.8%, and TXB2 decreased by 46.1%.

In the Les-6222 group the content of COX-1 decreased by 5.6% compared to the parameter of the control pathology group ($p < 0.01$), and COX-2—by 44.5% ($p < 0.01$). The content of PGE2 increased by 26.3% ($p < 0.01$) compared to the similar parameter of the control pathology group, the level of PGF2$\alpha$ decreased by 18.1% ($p < 0.01$), PGI2 increased by 27.2% ($p < 0.01$), and TXB2 decreased by 43.6% ($p < 0.01$).

In the model of chronic epileptogenesis, the content of 8-isoprostane, a marker of lipid peroxide oxidation, with a high degree of reliability increased by 2.78 times ($p < 0.01$) compared to the value of the intact control. This confirms the development of neuroinflammation under PTZ kindling conditions. Under the influence of sodium valproate, the level of 8-isoprostane decreased statistically significantly by 41.39% ($p < 0.01$) compared to the control pathology group. Celecoxib significantly reduced its content by 26.07% ($p < 0.01$) and the Les-6222 by 16.32% against the control pathology group ($p < 0.01$). This testifies to the antioxidant properties of all the tested compounds.

NSE is a biomarker that is associated with microglial activation. It makes it possible to assess the degree of brain neuron damage that occurs during ischemia and various metabolic, inflammatory, and neurodegenerative diseases [26]. A 17.5-fold increase in the level of NSE in the control pathology group compared to the intact control indicates the development of a rapid process of neuron impression in the brain of mice. In the sodium valproate group, the NSE level was statistically significantly reduced by 51.4% compared to the PTZ-induced kindling group, which is characteristic of this anticonvulsant [27]. No statistically significant decrease in this marker was found in the Celecoxib group, which indicates that the highly selective COX-2 blocker lacks distinct neuroprotective properties. Under the influence of Les-6222, the level of NSE was statistically reduced compared to the indicator of the control pathology group by 18.4%. This indicates a similar mechanism of action between Les-6222 and sodium valproate in counteracting PTZ-induced neuronal damage in models of chronic epileptogenesis.

Taking into account the high anti-inflammatory potential of 5-[(Z)-(4-nitrobenzylidene)]-2-(thiazol-2-ylimino)-4-thiazolidinone, it was reasonable to perform molecular docking studies on several anti-inflammatory targets. Initially, the structures of COX-1 (PDB code 4O1Z) and COX-2 (PDB code 3LN1) were selected for docking procedures as standard anti-inflammatory targets. Before docking, the 3D structure of Les-6222 was prepared using HyperChem 7.5 software. The MM+ molecular mechanic's method optimized the molecule's structure, achieving an RMS gradient of less than 0.1 kcal/(mol Å). The semiempirical quantum chemical method PM3 performed the final minimization of the energies of the studied intermediates until the RMS gradient was less than 0.01 kcal/(mol Å). However, the obtained binding energy values were small enough to explain the anti-inflammatory

activity of Les-6222 through COX inhibition. That is why the structures of 5-LOX (PDB code 3V99) and the FLAP enzyme (PDB code 6VGI) were also used for molecular docking.

The results of determining the binding energy of Les-6222 complexes and comparing drugs with inflammatory enzymes are given in Table 3.

**Table 3.** The binding energy of Les-6222 complexes and comparison drugs with biotargets.

|  | COX-1, kcal/mol | COX-2, kcal/mol | 5-LOX, kcal/mol | FLAP, kcal/mol |
|---|---|---|---|---|
| Les-6222 | −6.9 | −6.9 | −7.5 | −7.7 |
| Meloxicam | −9.8 | – | – | – |
| Celecoxib | – | −12.4 | – | – |
| Licofelon | – | – | −8.0 | – |
| MK-886 | – | – | – | −7.9 |

According to the obtained data, Les-6222 showed a high affinity for 5-LOX and FLAP and a lower binding energy for COX-1/2. Thus, 5-[(Z)-(4-nitrobenzylidene)]-2-(thiazol-2-ylimino)-4-thiazolidinone can be considered as a potential COX-1/COX-2/5-LOX/FLAP inhibitor. These results are essential for further in-depth study of the mechanisms of neuroinflammation. It is worth mentioning that the literature described similar studies about licofelone as a potent dual COX/LOX inhibitor, which has prominent anticonvulsant activity [28].

The 4-thiazolidinone derivative Les-6222 forms two hydrogen bonds with Arg596 and His367 with lengths of 2.85 Å and 2.12 Å, respectively. All three molecule cycles are connected to the number of lipophilic amino acids via the different types of hydrophobic non-covalent interactions of 5-LOX, as shown in Figure 2.

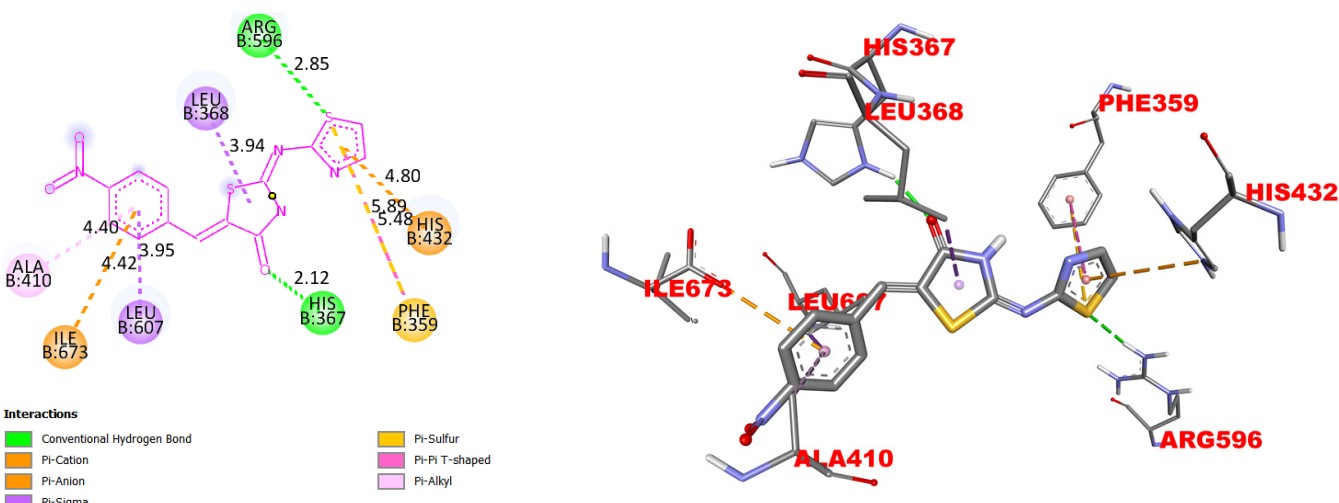

**Figure 2.** 2D and 3D binding mode of Les-6222 with 5-LOX.

According to molecular docking studies, Les-6222 has a pronounced affinity for FLAP with binding energy close to that of MK-886 (a leukotriene antagonist). The molecule does not form any hydrogen bonds and occupies a hydrophobic pocket formed by several lipophilic amino acids (Figure 3). The same interactions without any hydrogen bonds can be observed in the MK-886-FLAP complex.

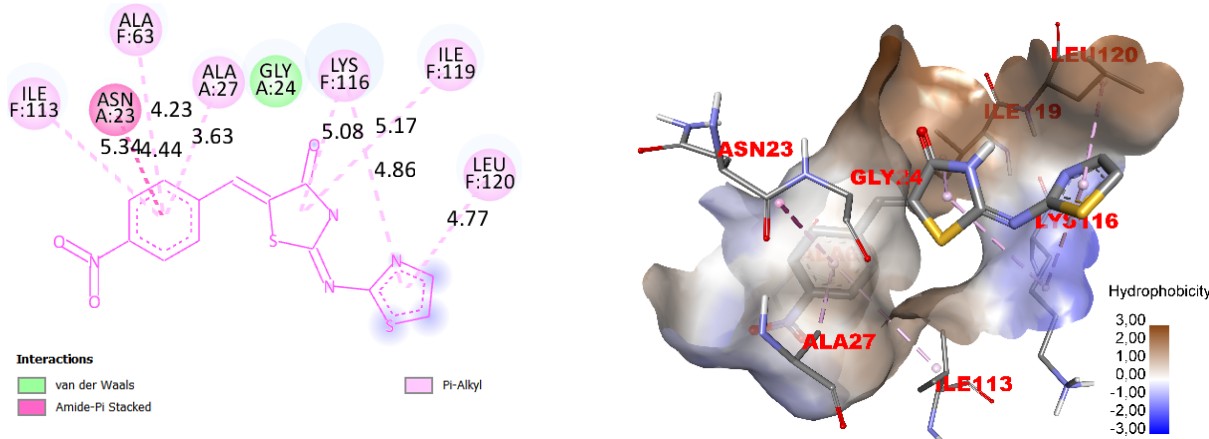

**Figure 3.** 2D and 3D binding mode of Les-6222 with FLAP.

Thus, the results of the conducted molecular docking study confirm the presence of affinity of 5-[(Z)-(4-nitrobenzylidene)]-2-(thiazol-2-ilimino)-4-thiazolidinone to the targets of inflammation and determine the expediency of further in-depth studies of the effect of the compound on the 5-lipoxygenase pathway of the arachidonic acid cascade.

## 4. Discussion

Our study showed that the course of PTZ-induced kindling in mice significantly depends on the applied experimental therapy. Under the influence of the classic anticonvulsant sodium valproate and the studied Les-6222, a pronounced anticonvulsant effect was observed, while Celecoxib did not improve the course of the model pathology. In our model of chronic epileptogenesis, the highly selective COX-2 inhibitor was clinically ineffective. The high level of NSE also confirms this in the mouse brain homogenate. Moderate anticonvulsant activity of Celecoxib in seizure models is known [8], which was confirmed in previous studies on a model of acute PTZ-induced seizure syndrome [29]. However, Celecoxib's protective properties are insufficient for effective seizure control in the kindling model.

COX-2 is an inducible enzyme in the inflammation process in response to various phlogogens. It is known that the expression of COX-2 increases in patients with epilepsy [30] and in animals during seizure modeling [31]. COX-2 catalyzes the conversion of arachidonic acid to the intermediate prostaglandin-H2, which is then converted by cell-specific synthases to thromboxane-A2 and four different prostaglandins: PGD2, PGE2, PGF2α, and PGI2, which are called prostanoids.

In our study on the PTZ-induced kindling model in the control pathology group, the content of COX-2 increased almost twice, which was not observed under the action of Les-6222 and sodium valproate. Compared to COX-2, the content of COX-1 only slightly increased in the model of chronic epileptogenesis. Therefore, the expression of COX-2 increases in the brain of mice in the background of PTZ kindling. Les-6222 preferentially inhibits COX-2, thus indicating its high selectivity for this enzyme, reducing the risk of side effects due to inhibiting constitutive COX-1. Interestingly, the tested 4-thiazolidinone derivative and its analogues in in vitro COX Inhibitor Screening Assay demonstrated predominantly COX-1 inhibition, nor COX-2 as described by Geronikaki et al. [32]. Sodium valproate also markedly inhibits COX-2.

According to the results of the correlation analysis, a moderate negative relationship is observed between COX-1 and COX-2 ($\rho = -0.43$). After PTZ kindling, a strong relationship ($\rho = 0.83$, $p < 0.05$) appears in the control pathology group, which indicates a direct dependence on the expression of both forms of COX. This dependence is weakened by the least effective anticonvulsant Celecoxib ($\rho = -0.31$). The highly effective agents Sodium Valproate and Les-6222 invert it, returning to the regularity characteristic of intact animals

($\rho = -0.60$ and $\rho = -0,66$, respectively), which indicates the involvement of the influence on the regulation of the arachidonic acid cascade.

Prostacyclin is an antagonist of TXB2. In intact animals, their contents have a weak correlation ($\rho = 0.31$). In the model of epileptogenesis, its strength increases ($\rho = 0.83$), which may indicate compensatory mechanisms of regulation of prostacyclin content and the processes of vasoconstriction and thrombus formation. On the background of anticonvulsant compound Les-6222 and sodium valproate, this relationship is inverted ($\rho = -0.54$ and $\rho = -0.43$, respectively), which can be considered as a marker of a common link in the mechanism of anticonvulsant and anti-inflammatory action of these agents, in contrast to celecoxib, which almost did not affect the correlation and left it at the level of control pathology ($\rho = 0.71$).

Celecoxib was found to not affect or target the brain of mice with a model of chronic epileptogenesis, but showed potent inhibition of COX-1. A study [8] reported the presence of possible non-selectivity of the COX-2 inhibitor Celecoxib, which can be explained by the use of high doses of the compound since its selectivity coefficient for COX-2 ($IC_{50} = 7.6$). It is also possible that such atypical nature of the effect of celecoxib on COX isoforms is due to the peculiarities of the model pathology.

In the study [33], to confirm the correlation between COX-2 level and PG production, PGE2, PGH2, PGD2, and PGI2 were determined in the hippocampus of intact mice on the model of PTZ-induced kindling, including under experimental celecoxib therapy. The authors showed that the content of PGE2 and PGH2 significantly increased in the hippocampus of mice under the influence of PTZ. Notwithstanding, celecoxib under these conditions significantly inhibited the formation of PGE2 and PGH2 and had almost no effect on PGD2 and PGI2. In our study using a similar model, the level of PGE2 in the homogenate of the whole brain, on the contrary, decreased almost by half, and against the background of the studied drugs, including celecoxib, with a close level to the intact control. The level of PGI2 in the mice in the control pathology group also decreased almost by half. This significant difference may be related to the fact that prostanoids were determined directly in the hippocampus of animals, where the main inflammatory processes of the epileptic brain are concentrated, and PGE2 regulates membrane excitability and long-term synaptic plasticity in perforant pathways [33].

In another study [34], the level of tumor necrosis factor-$\alpha$ (TNF-$\alpha$), interleukin-1$\beta$ (IL-1$\beta$), malondialdehyde (MDA) and PGE2, as well as glutathione (GSH) content was studied. In the homogenate of the brain, extracted 24 h after the last PTZ administration, a statistically significant increase in the level of PGE2 by 8 times was found in the group of control pathology compared to the level of intact animals. Under the influence of sodium valproate, the level of prostaglandin also increased, but significantly less than in the group of control pathology.

The shift in inflammatory mediators in several studies and in our experiment can be explained by the fact that the brain was removed 1 h after the last PTZ administration—much earlier than in previous works [33,34]. Therefore, such a discrepancy may indicate a certain phasic nature of changes in inflammatory mediators. Our approach to choosing a time point is based on numerous data in the literature, according to which the expression of COX-1, COX-2, and other mediators gradually increases with strengthening at the stage of seizure onset and within 1 h [35–37]. Twenty-four hours after seizures, the ratio of the mediators changes.

The role of individual prostaglandins in epileptogenesis is not clearly defined and can only be assessed comprehensively, taking into account all aspects of neuroinflammation. Experimental evidence suggests that PGE2 is a critical mediator in COX-2 signaling [38]. Moreover, administration of exogenous PGE2, but not PGD2 or PGF2$\alpha$, increases seizure frequency and amplitude of excitatory postsynaptic potentials [39]. In addition, PGE2 increases glutamate release from astrocytes [40], suggesting a role for PGE2 in controlling excitatory transmission in the brain. However, another study [41] showed that PGE2

protects cultured cortical neurons from NMDA-receptor-mediated glutamate neurotoxicity via EP2 receptors, i.e., exhibits neuroprotective properties.

The level of PGF2$\alpha$ increases in cerebral ischemia and epilepsy. This prostaglandin exerts its neurotoxic effect through PGF2$\alpha$ receptors coupled to a G protein [42]. In the context of these data, the decrease in PGF2$\alpha$ content in the brain suggests a protective effect of the studied Les-6222.

PGI2 exhibits neuroprotective properties in ischemic neuronal damage (including epilepsy) as it improves cerebral blood circulation [43]. In our study, the level of PGI2 was statistically significantly ($p < 0.01$) increased under the influence of Les-6222 and sodium valproate, which is evidence of a positive effect on neurons in a model of chronic epileptogenesis.

Prostacyclin antagonist TXB2 stimulates platelet aggregation and causes vasoconstriction, which is essential in developing epileptic brain ischemia [44]. Its decrease under the action of Les-6222 and sodium valproate can be considered a link to the neuroprotective activity of the studied compounds.

The molecular docking results showed Les-6222 inherent affinity for 5-LOX and FLAP and had weaker binding energy for COX-1/2. Activation of the 5-LOX inflammatory pathway is crucial in the neurodegenerative processes, which were widely described and discussed in the scientific literature [45–47]. Thus, the studied 4-thiazolidinone derivative can be considered as a potential inhibitor not only of COX-1/COX-2 but also of 5-LOX/FLAP, which provides background for further in-depth studies of the lipoxygenase link of the arachidonic acid cascade as a promising target in neuroinflammation.

Convulsive states contribute to the consumption of metabolic energy in the central nervous system and hypoxia of the central nervous system, which leads to the development of oxidative stress. The latter is one of the mechanisms involved in epileptic brain formation [48]. The high efficiency of the compound in reducing the content of 8-isoprostane is a favorable link in the mechanism of realizing the anticonvulsant effect.

According to the literature data, in diseases associated with the direct involvement of nervous tissue in the pathological process, qualitative and quantitative determinations of NSE indicate the severity of neuronal damage and violations of the general integrity of the blood–brain barrier [49].

As can be seen from Table 2, the levels of NSE in the Les-6222 and sodium valproate groups are statistically significantly ($p < 0.01$) lower than the level in the control pathology group. This proves the effectiveness of these drugs in anticonvulsant activity, which is confirmed by a significant ($p < 0.05$) decrease in the % of mice with seizures and a decrease in the number of days with paroxysms against the indicator of the control pathology group. Les-6222 and sodium valproate also produced a neuroprotective effect, protecting neurons from damage and death, and reducing inflammation in the brains of mice.

Thus, numerous inflammation mediators are involved in chronic epileptogenesis, and its mechanism requires further investigation. It is unclear whether blocking the neuroinflammatory pathway with COX inhibitors can prevent seizures under these conditions. The anticonvulsant properties of COX inhibitors and their role in the process of epileptogenesis require further targeted study. However, the promising anticonvulsant agent 5-[(Z)-(4-nitrobenzylidene)]-2-(thiazol-2-ylimino)-4-thiazolidinone, like the classical antiepileptic drug sodium valproate, exhibits pronounced anti-inflammatory properties in the brain of animals with a model of chronic epileptogenesis by inhibiting the cyclooxygenase pathway of the arachidonic acid cascade, which is associated with a reduction of oxidative stress and neuronal damage. Additionally, the tested 4-thiazolidinone derivative favorably differs from sodium valproate in the characteristics of secondary pharmacodynamics, in particular, in the absence of prodepressant side effects [50].

## 5. Conclusions

The results of the anticonvulsant activity of the promising anticonvulsant 5-[(Z)-(4-nitrobenzylidene)]-2-(thiazol-2-ylimino)-4-thiazolidinone on the model of PTZ-induced kindling with the determination of the effect on the content of inflammatory markers

(indicators of the arachidonic acid cascade COX-1, COX-2, PGE2, PGF2α, PGI2 and TXB2, as well as NSE in the homogenate of the mouse brain) are presented. In the model of chronic epileptogenesis, the studied 4-thiazolidinone derivative shows a pronounced anticonvulsant activity with marker indicators (latent period of attacks, percentage of animals with seizures, total number of days with seizures) comparable to sodium valproate. In the model of PTZ-induced kindling, the studied Les-6222, like sodium valproate, clearly affects the cyclooxygenase pathway of the arachidonic acid cascade, reducing the level of COX-1, COX-2, PGF2α and TXB2. The molecular docking confirmed that the compound Les-6222 has anti-inflammatory properties and affinity for 5-LOX and FLAP inflammatory targets. Les-6222 and sodium valproate exhibit neuroprotective properties and decrease the level of NSE by 18.4 and 51.4% in animal brain homogenate.

**Author Contributions:** Conceptualization, S.S. and R.L.; methodology and experimental work, M.M., M.H., D.K. and T.G.; data analysis, S.S., A.L. and R.L.; writing—review and editing, M.M., S.S., A.L., D.K. and R.L.; project administration and supervision, S.S. and R.L. All authors have read and agreed to the published version of the manuscript.

**Funding:** The research leading to these results has received funding from the Ministry of Health of Ukraine, under the project numbers: 0121U100690, 0120U102460, and the National Research Foundation of Ukraine, under the project number: 2020.02/0035.

**Institutional Review Board Statement:** The study was conducted according to the guidelines of the Declaration of Helsinki and approved by the Ethical Committee of the National University of Pharmacy, Ukraine (protocol No. 3 from 20 March 2019).

**Informed Consent Statement:** Not applicable.

**Data Availability Statement:** The data presented in this study are available in this article.

**Acknowledgments:** The authors would like to thank all the brave defenders of Ukraine who made the finalization of this article possible.

**Conflicts of Interest:** The authors declare no conflict of interest.

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
