# Peer review of "Evaluation of 5-[(Z)-(4-nitrobenzylidene)]-2-(thiazol-2-ylimino)-4-thiazolidinone (Les-6222) as Potential Anticonvulsant Agent"

_scipharm, doi:10.3390/scipharm90030056_

Round 1

Reviewer 1 Report

Title: Evaluation of 5-[(Z)-(4-nitrobenzylydene)]-2-(thiazol-2-ylimino)-4-thiazolidinone as potential anticonvulsant.

Authors: Mariia Mishchenko, Sergiy Shtrygol, Andrii Lozynskyi, Mykhailo Hoidyk, Dmytro Khyluk, Tatyana Gor-bach and Roman Lesyk.

Lesyk et. al.'s work is clearly straightforward and easy to grasp. The authors have carried out a significant work towards the development of new anticonvulsant agents for the treatment of epilepsy. While going through the detailed study of the work the following questions or remarks came to my mind. Would you please comment on them?

- I would recommend changing the title to be ‘Evaluation of 5-[(Z)-(4-nitrobenzylydene)]-2-(thiazol-2-ylimino)-4-thiazolidinone (Les-6222) as potential anticonvulsant agent’.

- In PTZ-induced kindling model, you mentioned that you administered the drugs intragastrically 30 minutes before the administration of pentylenetetrazol, is 30 minutes enough for the onset of their actions? and what is the difference in half-lives between the three investigated drugs.

- Could you comment why the level of the COX-2 was not affected by celecoxib (the selective COX-2 inhibitor), however, 52% and 45% decrease was observed with sodium valproate and Les-6222 respectively.

- Have you investigated the physicochemical properties of Les-6222. Optimal solubility and blood brain barrier permeability will be required for further in vitro and in vivo studies. Also, have you observed any toxicities with Les-6222, since it has a nitro containing compound.

- In your previous study (Sci. Pharm. 2020, 88, 16), you investigated also carbamazepine as a reference anticonvulsant drug, in addition to sodium valproate. In this study you did not use carbamazepine, would you please comment on that?

In line 26, Is it COX or COX-2

- In line 96, I guess there is an extra ‘a’, it could be ‘we designed and synthesized 5-[(Z)-(4-nitrobenzylidene)]-…… ‘

- In Line 100, I would recommend changing the figure description to ‘Figure 1. Chemical structure of compound Les-6222.’

In line 292, you missed to mention on the text that the described interactions of Les-6222 are with 5-LOX as shown in figure 2.

Author Response

Dear reviewer!

Many thanks for Your time spending and efforts to review the manuscript. Your suggestions have been incorporated in the revised manuscript (yellow highlight).

We would like to comment the main points.

“- I would recommend changing the title to be ‘Evaluation of 5-[(Z)-(4-nitrobenzylydene)]-2-(thiazol-2-ylimino)-4-thiazolidinone (Les-6222) as potential anticonvulsant agent’.”

We changed the title of the manuscript according to Your notes.

“- In PTZ-induced kindling model, you mentioned that you administered the drugs intragastrically 30 minutes before the administration of pentylenetetrazol, is 30 minutes enough for the onset of their actions? and what is the difference in half-lives between the three investigated drugs.”

Yes, such a protocol meets methodological recommendations. This time is quite enough, as evidenced by the long research experience, including those published in our previous articles. He confirms that an anticonvulsant effect develops during this time. Pharmacokinetic features of the Les-6222 compound, including half-life, were not the subject of this study. They require a separate study.

“- Could you comment why the level of the COX-2 was not affected by celecoxib (the selective COX-2 inhibitor), however, 52% and 45% decrease was observed with sodium valproate and Les-6222 respectively”

In the conditions of kindling, the activation of COX (including the inducible isoform of COX-2) is the result of damage to neurons due to the long-term action of the proconvulsant. Sodium valproate and Les-6222, due to their pronounced anticonvulsant effects, appear to be better at preventing this initial brain damage that triggers the inflammatory response.

“- Have you investigated the physicochemical properties of Les-6222. Optimal solubility and blood brain barrier permeability will be required for further in vitro and in vivo studies. Also, have you observed any toxicities with Les-6222, since it has a nitro containing compound”

The physicochemical properties, solubility, blood-brain barrier permeability and toxicity of Les-6222 and their structure-related analogues were investigated in our previous reports [Ref. 19, 21] and other authors [Ref 32] and references are cited in the current manuscript.

  1. Mishchenko, M.V.; Shtrygol, S.Yu. Spectrum of anticonvulsant activity and acute toxicity of 5-[(Z)-(4-nitro-benzylidene)]-2-(thiazol-2-ilimino)-4-thiazolidinone. Pharmacology and medicinal toxicology 2020, 14, 389-396. (in Ukrainian) https://doi.org/10.33250/14.06.389
  2. Mishchenko, M.; Shtrygol, S.; Lozynskyi, A.; Khomyak, S.; Novikov, V.; Karpenko, O.; Holota, S.; Lesyk, R. Evaluation of anticonvulsant activity of dual COX-2/5-LOX inhibitor darbufelon and its novel analogues. Sci. Pharm. 2021, 89, 22. https://doi.org/10.3390/scipharm89020022
  1. Geronikaki, A.A.; Lagunin, A.A.; Hadjipavlou-Litina, D.I.; Eleftheriou, P.T., Filimonov, D.A.; Poroikov, V.V., Alam, I.; Saxena, A. K. (2008). Computer-aided discovery of anti-inflammatory thiazolidinones with dual cyclooxygenase/lipoxygenase inhibition. J. Med. Chem. 2008, 51, 1601-1609. https://doi.org/10.1021/jm701496h.

“- In your previous study (Sci. Pharm. 2020, 88, 16), you investigated also carbamazepine as a reference anticonvulsant drug, in addition to sodium valproate. In this study you did not use carbamazepine, would you please comment on that?”

Thus, in the study (Sci. Pharm. 2020, 88, 16), we used carbamazepine, but in the MES test, since the dominant mechanism of action of carbamazepine (blockade of sodium channels) corresponds well to the pathogenesis of the specified model. In the PTZ-induced kindling model, the pathogenesis of which is mainly associated with the suppression of GABA-ergic inhibitory processes, it was considered inappropriate to use carbamazepine in view of its mechanism of action. For these reasons, the optimal comparator for this model is sodium valproate.

In addition, we revised carefully the manuscript language, changed the Figure 1 description according to Your notes and hope the current version it will acceptable for publication.

Sincerely Yours,

Prof. Roman Lesyk

Reviewer 2 Report

In the article entitled "Evaluation of 2 5-[(Z)-(4-nitrobenzylydene)]-2-(thiazol-2-ylimino)-4-thiazolidinone as potential anticonvulsant", interesting results are observed, which should be better focused taking into account that there is information related to the anti-inflammatory properties of this compound, however, in this work they are extended to other tests, which can be better exploited when comparing the previous results reported with those obtained. This would allow reaching more solid and grounded conclusions. I suggest a review of scientific literature to enrich the writing, as well as a better distribution of information, especially in the discussion item. For specific aspects in some items, I send the following questions and suggestions:

Abstract:

-Why can it be said that there is high selectivity of Les-6222 for the COX-2 enzyme? it is important to clarify based on the results

4. Discussion:

-Could the results of the inflammatory markers COX-1, COX-2, PGE2, PGF2α, PGI2 and TXB2 be related to the anti-inflammatory results of some type of analog? It would be interesting to share, since it is not the only result of this compound against anti-inflammatory activity. It is important to take into account the article doi 10.1021/jm701496h (which has more than 100 citations), which mentions the prediction made by computer through docking studies of compounds such as the one mentioned in this article, which has anti-inflammatory activity due to dual inhibition of cyclooxygenase/lipoxygenase (COX/LOX) . Is there a correlation with the results reported in this article and what differences does it present?

-According to the docking study carried out, can similarity be found in the mode of action of sodium valproate with Les-6222? Or since their chemical structure is different, could they have different activation pathways? Try to highlight the results found based on others reported for this same type of compound.

-Greater organization of the information is required, that it be better exposed and that there be comparative tables with information previously reported in other articles, since the exposition of the information in this item is unclear.

5. Conclusions:

-It is important in this item to show the most important results, if in the discussion it is stated why there is a greater selectivity to COX-2 or if the NSE values are significant, they must refer and give the reason for their importance. Do not be so general in the conclusions of the article

Author Response

Dear reviewer!

Many thanks for Your time spending and efforts to review the manuscript. Your suggestions have been incorporated in the revised manuscript (yellow highlight).

We would like to comment the main points.

“-Why can it be said that there is high selectivity of Les-6222 for the COX-2 enzyme? it is important to clarify based on the results.”

This is based on the fact that Les-6222 preferentially reduces COX-2 (by 44.5%, while COX-1 is only reduced by 5.6%) and we insert the mentioned information according to Your notes.

“Could the results of the inflammatory markers COX-1, COX-2, PGE2, PGF2α, PGI2 and TXB2 be related to the anti-inflammatory results of some type of analog? It would be interesting to share, since it is not the only result of this compound against anti-inflammatory activity. It is important to take into account the article doi 10.1021/jm701496h (which has more than 100 citations), which mentions the prediction made by computer through docking studies of compounds such as the one mentioned in this article, which has anti-inflammatory activity due to dual inhibition of cyclooxygenase/lipoxygenase (COX/LOX). Is there a correlation with the results reported in this article and what differences does it present?”

Thiazolidine derivatives have anti-inflammatory properties. The results of computer prediction and elucidation of anti-inflammatory activity in vitro and in vivo of these derivatives, including those mentioned in this article [Athina A. Geronikaki et al., Computer-Aided Discovery of Anti-Inflammatory Thiazolidinones with Dual Cyclooxygenase/ Lipoxygenase Inhibition. J. Med. Chem. 2008, 51, 1601–1609]. However, our study evaluated the anti-inflammatory properties for the first time in the context of the connection between seizure syndrome and neuroinflammation. According to [Athina A. Geronikaki et al., Computer-Aided Discovery of Anti-Inflammatory Thiazolidinones with Dual Cyclooxygenase/Lipoxygenase Inhibition. J. Med. Chem. 2008, 51, 1601–1609 ], the studied compounds inhibit COX-1 (8-90%) to a greater extent than COX-2 (0-30%), and also moderately inhibit LOG (12-76%). Our results confirm the anti-inflammatory properties of compounds of the studied series. But the difference is that in the conditions of the neuroinflammation model of chronic epileptogenesis when COX-2 is mainly activated in the brain, the compound Les-6222 in vivo suppresses COX-2 and significantly less COX-1. The docking results indicate the promise of further in vivo study of the effect of Les-6222 on the LOX-dependent inflammatory pathway, as this compound demonstrated a high affinity for 5-LOX and FLAP. The results of mentioned article and reference we insert in manuscript according to Your notes.

“-According to the docking study carried out, can similarity be found in the mode of action of sodium valproate with Les-6222? Or since their chemical structure is different, could they have different activation pathways? Try to highlight the results found based on others reported for this same type of compound”.

According to literature data, sodium valproate expresses its activity by the influence on the GABAergic system and the effect on enzymes like succinate semialdehyde dehydrogenase (SSA-DH), GABA transaminase (GABA-T), and α-ketoglutarate dehydrogenase. But still, the mechanisms of action of the sodium valproate remain unknown [1]. In addition, there is the absence of any spectrum in the Protein Data Bank, which has the valproate as a co-crystalized ligand. Docking studies confirm the proposed pharmacodynamical profile of the Les-6222 as COXs/5-LOX inhibitors. Such inhibition gives results in anticonvulsant activity, which is also confirmed in other research, especially for licofelone as a dual COX/LOX inhibitor with anticonvulsant activity [2]. The mentioned information we added in the current manuscript. The results of mentioned article and reference we insert in the manuscript according to Your notes.

  1. Johannessen, Cecilie U. "Mechanisms of action of valproate: a commentatory." Neurochemistry international 37.2-3 (2000): 103-110.
  2. Payandemehr, Borna, et al. "A COX/5-LOX inhibitor licofelone revealed anticonvulsant properties through iNOS diminution in mice." Neurochemical research 40.9 (2015): 1819-1828.

-Greater organization of the information is required, that it be better exposed and that there be comparative tables with information previously reported in other articles, since the exposition of the information in this item is unclear.”

We partly added previously reported data in the Discussion section according to Your suggestions.

-It is important in this item to show the most important results, if in the discussion it is stated why there is a greater selectivity to COX-2 or if the NSE values are significant, they must refer and give the reason for their importance. Do not be so general in the conclusions of the article”

We partly insert some significant result values in the Conclusion section according to Your suggestions and also revised carefully the manuscript language and hope in the current version it will acceptable for publication.

Sincerely Yours,

Prof. Roman Lesyk

Reviewer 3 Report

Mishchenko et al. investigated a 4-thiazolidinone compound (Les-6222)as a potential anticonvulsant in comparison with sodium valproate and celecoxib in mice model of chronic epileptogenesis. Les-6222 showed similar anti-convulsant effect to the classic sodium valproate in the pentylenetetrazole induced mouse model. In addition the authors also investigated the effect of Les-6222, sodium Valproate, and Celecoxib on the cyclooxygenase pathway of the arachidonic acid cascade as well as 8-isoprostane and Neuron specific enolase levels in the mouse model. It was shown that Les-6222 affects this pathway by reducing the level of COX-1, COX-2, PGF2alpha and TXB2. Molecular docking suggested the Les-6222 could have affinity for COX-1, COX-2, 5-LOX, and FLAP proteins.

Overall the manuscript is well written and the generated data support the conclusions. Of course it could be argued that the modeling data should be backed up by proper binding assay experiments to determine the binding constants etc. It does not seem clear for the general reader as to why or how Les-6222 is an improvement over the classical anti-convulsant sodium Valproate. Some explanation regarding the molecular docking results would be helpful to the reader as well, such as if there's any homology between the COX enzymes and 5-LOX and FLAP? why does Les-6222 have affinity for all 4?  The authors might also consider mentioning if  5-LOX and FLAP have any significance in the disease.

The authors state that Les-6222 and sodium valproate share a similar mechanism of action (line 265-) but do not explain any further what the mechanism is. Does sodium valproate have any affinity for COX-1, COX-2, 5-LOX or FLAP?

The authors should address/discuss/comment on that sodium valproate is just as effective as Les-6222. What is the advantage of using Les-6222 over sodium valproate? Is the goal to further develop Les-6222 to a more effective anti-convulsant?

Specific comment:

Line 239: 'the level of COX-1 decreased by 149%'- this should be 14.9%

Author Response

Dear reviewer!

Many thanks for Your time spending and efforts to review the manuscript. Your suggestions have been incorporated in the revised manuscript (yellow highlight).

We would like to comment the main points.

“It does not seem clear for the general reader as to why or how Les-6222 is an improvement over the classical anti-convulsant sodium Valproate. Some explanation regarding the molecular docking results would be helpful to the reader as well, such as if there's any homology between the COX enzymes and 5-LOX and FLAP? why does Les-6222 have affinity for all 4?  The authors might also consider mentioning if  5-LOX and FLAP have any significance in the disease. ”

Often some compounds, including 4-thiazolidinone derivatives possess an affinity to different enzymes [1]. Firstly, we forecasted the anti-inflammatory activity of 4-thiazolidinone derivative Les-6222 by inhibitions only COXs family enzymes because there are literally reported, namely 5-Arylidene-2-imino-4-thiazolidinones as the COX-2 inhibitors [2]. Nevertheless, the binding energies of the Les-6222-COXs complexes don’t allow to explain the effect of the compound only by the COXs inhibition. Therefore, we tested also 5-LOX and FLAP for possible affinities. Docking results allow the suggestion of the additional 5-LOX and FLAP pathways because activation 5-LOX inflammatory pathway is tremendously crucial in the neurodegenerative processes, which were widely described and discussed in the scientific literature [3-5]. We also observed similar studies about dual COX/LOX inhibitor licofelone with prominent anticonvulsant activity [6,7]. The mentioned information and references we added in the manuscript according to Your notes.

  1. Verma, Amit, and Shailendra K. Saraf. "4-Thiazolidinone–A biologically active scaffold." European journal of medicinal chemistry 43.5 (2008): 897-905.
  2. Ottana, Rosaria, et al. "5-Arylidene-2-imino-4-thiazolidinones: design and synthesis of novel anti-inflammatory agents." Bioorganic & medicinal chemistry 13.13 (2005): 4243-4252.
  3. Manev, Hari, et al. "Putative role of neuronal 5‐lipoxygenase in an aging brain." The FASEB Journal 14.10 (2000): 1464-1469.
  4. Razavi, Seyed Mehrad, et al. "Licofelone, a potent COX/5-LOX inhibitor and a novel option for treatment of neurological disorders." Prostaglandins & Other Lipid Mediators 157 (2021): 106587.
  5. Qu, Tingyu, Tolga Uz, and Hari Manev. "Inflammatory 5-LOX mRNA and protein are increased in brain of aging rats." Neurobiology of aging 21.5 (2000): 647-652.
  6. Fischer, L., et al. "The molecular mechanism of the inhibition by licofelone of the biosynthesis of 5‐lipoxygenase products." British journal of pharmacology 152.4 (2007): 471-480.
  7. Payandemehr, Borna, et al. "A COX/5-LOX inhibitor licofelone revealed anticonvulsant properties through iNOS diminution in mice." Neurochemical research 40.9 (2015): 1819-1828.

“The authors state that Les-6222 and sodium valproate share a similar mechanism of action (line 265-) but do not explain any further what the mechanism is. Does sodium valproate have any affinity for COX-1, COX-2, 5-LOX or FLAP?”

The claim of a similar mechanism of action in the context of the discussed fragment of the manuscript is based on the fact that both Les-6222 and especially sodium valproate reduces the content of NSE, a marker of neuronal damage. This means that the studied compound and the classic anticonvulsant have a common property – neurocytoprotective activity, which is important for anticonvulsant action. The affinity of valproate for COX-1, COX-2, 5-LOX, or FLAP was not determined in this study.

“The authors should address/discuss/comment on that sodium valproate is just as effective as Les-6222. What is the advantage of using Les-6222 over sodium valproate? Is the goal to further develop Les-6222 to a more effective anti-convulsant?”

In terms of anticonvulsant activity in the PTZ-induced kindling model, compound Les-6222 is not inferior to sodium valproate (Table 1). It also has a pronounced anti-inflammatory activity (it is not inferior to diclofenac sodium in the model of carrageenan edema) as observed in our previous reports. One of the advantages of Les-6222 over sodium valproate is that the test compound has low toxicity (Hodge and Sterner class V toxicity, since the LD50 when administered i.v. to mice exceeds 5000 mg/kg) [Ref 19. of our manuscript]. During a two-week observation of animals administered the compound Les-6222 at a dose of 5000 mg/kg, dyspeptic phenomena, blepharoptosis, respiratory depression and other adverse manifestations were not observed [Ref 19. of our manuscript]. For comparison, it is worth noting that the LD50 of sodium valproate when intravenously administered to mice is 977 mg/kg [Material Safety Data Sheet [Electronic resource]. 2010. https://datasheets.scbt.com/sc-202378.pdf (accessed on 02/20/2022). Valproic Acid, Sodium Salt. 296. Material Safety], Also, Les-6222 favorably differs from sodium valproate in the characteristics of secondary pharmacodynamics, in particular, in the absence of a prodepressant side effects [Ref. 50 of our manuscript]. All of this determines the prospects for further development of Les-6222 intending to create an effective anti-epileptic drug. The mentioned information we added according to Your suggestions.

In addition, we revised carefully the manuscript text, and hope the current version it will acceptable for publication.

Sincerely Yours,

Prof. Roman Lesyk